# Tooth-Colored CAD/CAM Materials for Application in 3-Unit Fixed Dental Prostheses in the Molar Area: An Illustrated Clinical Comparison

**DOI:** 10.3390/ma13245588

**Published:** 2020-12-08

**Authors:** Angelika Rauch, Sebastian Hahnel, Elena Günther, Wolfgang Bidmon, Oliver Schierz

**Affiliations:** Department of Prosthodontics and Materials Science, University of Leipzig, Liebigstr. 10, Haus 1, 04103 Leipzig, Germany; sebastian.hahnel@medizin.uni-leipzig.de (S.H.); elena.guenther@medizin.uni-leipzig.de (E.G.); wolfgang.bidmon@medizin.uni-leipzig.de (W.B.); oliver.schierz@medizin.uni-leipzig.de (O.S.)

**Keywords:** clinical evaluation, dental prosthesis, fixed partial denture, monolithic zirconia, polymers

## Abstract

The aim of this study was to compare the clinical properties of tooth-colored computer-aided design/computer-aided manufacturing (CAD/CAM) materials for the fabrication of a 3-unit fixed dental prostheses (FDPs) in the same clinical scenario. A 53-year-old female patient was supplied with a 3-unit FDP to replace a second premolar in the upper jaw. Restorations were fabricated from 3 mol%, 4 mol%, and 5 mol% yttrium oxide zirconia, zirconia with translucency gradient, indirect composite resin, polyetheretherketone (PEEK), and polyetherketoneketone (PEKK). Milling time, weight, and radiopacity were investigated. Esthetics were examined following the US Public Health Service criteria (USPHS). The milling time for zirconia was twice as high as for the indirect composite resin, PEEK, or PEKK. The latter materials had a weight of 2 g each, while zirconia restorations yielded 5 g. Zirconia presented intense radiopacity. PEEK and PEKK required veneering and an opaquer was applied to the PEKK framework. All FDPs showed acceptable esthetics. PEEK and PEKK restorations were featured by a grayish shimmering. A variety of CAD/CAM materials are available to fabricate 3-unit FDPs with esthetically acceptable results. In the esthetic zone, PEEK and PEKK require veneering and an opaquer might be applied. Milling time, weight, and radiopacity were relatively high for zirconia FDPs.

## 1. Introduction

The conventional wisdom in restorative dentistry is that the esthetic appearance of dental restorations has gained increasing importance in the last decades, which is why the application of tooth-colored materials is highly favored by both patients and dentists. Recent surveys revealed that dentists often recommend ceramics, especially lithium disilicate and zirconia, for anterior and posterior single crowns [1,2]. However, if it comes to multi-unit fixed dental prostheses (FDPs), the range of materials is limited, since most of the silica-based ceramics are not approved for the fabrication of multi-unit FDPs. Pressed zirconia-reinforced lithium silicate ceramics, and milled computer-aided designed/computer-aided manufactured (CAD/CAM) or pressed lithium disilicate ceramics are available for the fabrication of 3-unit FDPs, yet only up to the second premolar. In the molar area, none of the silica-based ceramics are recommended for the application in multi-unit FDPs. For these scenarios, ceramics with a polycrystalline structure are available, especially zirconia, which has been extensively used during the last years. The formulation of zirconia has been modified regarding its yttrium oxide dotation. Thus, different zirconia materials are available on the dental market and the amount of yttrium oxide relevantly influences the mechanical and optical properties [3]. While zirconia restorations with 3 mol% yttrium oxide (3 Y-TZP) present high flexural strengths, an endowment to 5 mol% yields lower values. The latter formulation is not recommended for the fabrication of FDPs with more than three units, but improved translucency may coincide with improved esthetics [4,5]. Indirect composite resins (CAD/CAM composite resins) are mostly recommended for partial restorations or single crowns. In comparison to ceramics, they are featured by high edge stability, lower brittleness, and a dentine-like elastic modulus [6]. Two composite resins (LuxaCam Composite, DMG, Hamburg, Germany; Trinia, SHOFU, Kyoto, Japan) are available for application in multi-unit FDPs [6]. In the recent years, polyetheretherketone (PEEK) and polyetherketoneketone (PEKK), which belong to the superordinate group of polyaryletherketone (PAEK), have been increasingly promoted for application in fixed prosthodontics. PAEK materials feature high biocompatibility and favorable mechanical properties. They are available in colors similar to that of teeth, e.g., dentine, whitish, universal, and in other colors such as grayish or pink [7,8,9,10].

Tooth-colored materials currently available on the market for the fabrication of 3-unit FPDs belong to different classes regarding their chemical structure and composition. It is therefore clear that their distinct properties may impact the individual esthetic appearance of the restoration. In addition, it has been highlighted that zirconia restorations may produce artifacts in X-ray examinations [11], which indicates that the radiopacity of the restorative material can be a relevant issue for later diagnostic procedures. In contrast, for resin-based composites, radiopacity has been described as a desirable property as it allows the differentiation of carious lesions below the crown margin or decalcified dentin from the restoration as well as the location of voids or other defects within the restoration [12].

The differences in chemical structure and composition of the materials available on the market account for differences in the individual material’s density. With regard to this aspect, densities of 6 g/cm^3^ have been reported for zirconia, while PAEK and composite resin materials feature far lower densities between 1.3 and 2.0 g/cm^3^ [8,13,14]. From a clinical point of view, lower density of a restorative material results in less weight of the restoration, which might be favorable in patients requiring extensive restorations. Tooth-colored CAD/CAM materials are processed by using CAD software and can be milled with 4- or 5-axis CAM machines. The density of a CAD/CAM-processed restorative material could be a parameter that impacts milling efficiency, which suggests that denser restorative materials might require higher milling times.

Against this background, this clinical report illustrates the replacement of a second premolar by using a 3-unit fixed dental prosthesis fabricated from different restorative materials available on the market for this particular setting, comparing clinically relevant properties such as esthetics, radiopacity, and weight as well as parameters potentially relevant for the dental laboratory such as milling time. The working hypothesis was that there are no differences between the various materials and restorations regarding the properties investigated.

## 2. Materials and Methods

A 53-year-old healthy female patient introduced herself at the Department of Prosthodontics and Materials Science at Leipzig University (Leipzig, Germany). The second premolar in the left upper jaw was missing. The patient history revealed that she was supplied with a dental implant that had to be explanted a year before the appointment, due to an inflammation. Therefore, she was looking for an alternative to replace the missing premolar.

During clinical examination, healthy periodontal conditions were observed and a screening for temporomandibular disorders was negative. The patient denied being aware of bruxism, and during the clinical examination no signs of bruxism were identified. All teeth were naturally healthy or restored with sufficient restorations, except for a carious lesion in the left first molar of the upper jaw (Figure 1a,b).

As the patient was not interested in any implant-supported restoration, the fabrication of a 3-unit FDP was proposed to replace the missing tooth. The patient agreed to the concept and gave her signed consent to be part of the clinical trial. For comparisons, 3 Y-TZP, 4 Y-TZP, and 5 Y-TZP supplied by the same manufacturer were chosen, and another zirconia with translucency gradient, one composite resin, one PEEK, and one PEKK material were selected (Table 1).

The abutment teeth were prepared according to preparation guidelines for ceramics (Figure 2a and Figure 3a) and an impression was taken using the sandwich technique (Honigum heavy and light, DMG).

The tooth color was D3, which was determined by using the Vita classical shade guide (Vita Zahnfarbrik, Bad Säckingen, Germany). Working casts were prepared and digitalized with a laboratory scanner (inEos X5, Dentsply Sirona, Charlotte, NC, USA). Afterwards, the FDPs were designed (ceramill mind 2.4, Amann Girrbach, Pforzheim, Germany) in accordance with the manufacturer’s instructions (Table 1) and milled (inLab MC X5, Dentsply Sirona). The approximated milling time issued by the CAM software (inLab CAM SW 20.0.1, Dentsply Sirona) for the individual restoration was recorded. The 3 Y-TZP zirconia material was only available as white material from the manufacturer and needed to be colored with coloring liquids (Figure 4).

The FDPs fabricated from PEEK and PEKK materials were facially veneered with a composite resin using a conventional technique. For PEKK, an opaquer was priorly applied to mask the grayish framework.

To keep tooth hard tissue removal to the necessary minimum and to better compare the original colors, the occlusal and oral sides of all FDPs were fabricated in a monolithic approach. Only PEEK and PEKK restorations were facially veneered. All materials were characterized with stains to imitate the adjacent teeth (Table 2).

The weight of the FDPs was measured (PCB 1000-2, Kern & Sohn, Balingen-Frommern, Germany). An X-ray was made (7 mA, 70 kV, 40 ms; Heliodent Plus, Dentsply Sirona) to evaluate radiopacity with gray values (ImageJ 1.52p, National Institutes of Health, Bethesda, MD, USA) of the dentine of a reference abutment tooth as well as three distinct locations of the individual restorations (central in the pontic area, mesial at the premolar margin, and distal at the molar margin). An intraoral try-in of the FDPs was performed (Variolink Esthetic Try-In neutral, Ivoclar Vivadent, Schaan, Liechtenstein) and the patient was asked to choose her favorite FDP. Finally, all FDPs were evaluated for the parameters ‘anatomic form’, ‘marginal adaption’, and ‘color match’ according to USPHS guidelines [15] by an independent and trained examiner.

## 3. Results

The approximated milling time for zirconia with translucency gradient was highest, while half the time was estimated to mill FDPs from PEEK, PEKK, and the indirect composite resin (Table 2). The PEKK material that was labeled as whitish by its manufacturer appeared to be grayish and did not meet esthetic requirements by using only the veneering material. Therefore, it was covered with an opaquer beforehand. The weight of the restorations ranged between 5.0 and 5.3 g for zirconia and between 1.5 and 1.9 g for the other FDPs (Table 2). The X-ray examination of the reference abutment tooth revealed a mean gray value of 114 for dentine near the preparation margin (Figure 5, blue rectangle).

Zirconia restorations presented intense radiographic opacity with highest gray values of 255 in all points of interest (Figure 5, Table 2).

According to US Public Health Service (USPHS) criteria, the clinical evaluation of all FDPs revealed rankings of Alpha for the categories ‘anatomic form’ and ‘marginal adaption’ (Figure 2 and Figure 3). From the vestibular view, ‘color match’ of FDPs fabricated from PEEK and PEKK was rated as Bravo, since a relatively opaque and grayish shade was observed (Figure 3). The colors of restorations fabricated from zirconia and CAD/CAM composite resin were categorized as Alpha. The patient favored the zirconia FDP with translucency gradient, which was self-adhesively cemented (RelyX Unicem, 3M, St. Paul, MN, USA). 

## 4. Discussion

Regarding the esthetic appearance of the 3-unit FDPs investigated in the current clinical report, the results suggest that a variety of CAD/CAM materials can be employed to supply patients with esthetically favorable 3-unit FDPs in the posterior area. However, PEEK and PEKK frameworks should be covered with an opaquer and require veneering with a composite resin or ceramic material to produce acceptable esthetic results. Moreover, the materials featured differences in their radiopacity, with the lowest radiopacity were identified for PAEK materials and highest radiopacity for zirconia. Regarding laboratory processing, approximated milling time was different between the various materials. Thus, the working hypothesis of this clinical report was rejected.

This clinical report illustrates that acceptable esthetical results in the posterior area can be achieved with various formulation of zirconia. Composite resins yielded similar favorable esthetic outcomes, yet for PAEK materials, veneering seems necessary in the esthetic zone. Even if PAEK materials are available in colors that are described as whitish or universal by the manufacturer, these are not similar to those of natural teeth. However, for the veneering of PEEK with CAD/CAM composite resins or ceramics, similar colorimetric properties might be achieved than for zirconia or metal frameworks [16]. Besides the CAD/CAM technique for veneering, PAEK materials can be veneered by using conventional techniques or with pre-fabricated composite resin veneers. Nonetheless, the veneering of PAEK materials is challenging. In vitro results revealed that conventionally fabricated or pre-fabricated composite resin veneers showed lower fracture loads than those milled with the CAD/CAM technique and adhesive failure was high for pre-fabricated veneers [17]. The technique of bonding composite resins to PAEK is controversially discussed. A systematic review concluded that restorations should be pretreated with mechanical and/or chemical methods, such as air abrasion or sulfuric acid etching. Finally, a bonding agent containing methyl methacrylate (MMA) or 10-methacryloyloxydecyl dihydrogen phosphate (MDP) should be applied [11].

Regarding cementation, it is possible that restorations fabricated from zirconia ceramics, PEEK, and PEKK are conventionally cemented. This might be beneficial in comparison to restorations fabricated from CAD/CAM composite resins that need to be adhesively cemented. Nonetheless, debonding of composite resins is still a frequently described complication; thus, the adhesive cementation technique should be processed with caution [6].

The mechanical properties of CAD/CAM composite resins, PEEK, and PEKK are featured by a relatively low elastic modulus of 3 to 5 GPa. This value is close to that of dentine (15 GPa), which might yield a cushioning effect on tooth-supported restorations and might also be interesting in implant dentistry. Stress transferred to the abutment tooth or abutment/implant might be reduced in comparison to restorations fabricated from other materials such as zirconia with 200 GPa [8,18]. Due to the low density of applied composite resins and PAEKs in the present clinical report, these restorations yielded lighter weight than restorations fabricated from zirconia, since volumes of the FDPs were similar. A decreased weight of restorations might be beneficial for patient satisfaction in cases with extensive restorative reconstructions, for instance in full-arch restorations [19]. Moreover, the low density can be one parameter that has an impact on milling time in the CAM machines. In the present clinical report, the low density of resin composites and PAEK materials correlated with shorter approximated milling times, which might be an advantage for the chairside application of, e.g., CAD/CAM composite resins.

In X-ray examinations, margins of the FDPs fabricated from PAEK and composite resins were less radiopaque than dentine. Radiopacity of zirconia was most intense, which corroborates previous investigations [11]. Judging from these observations, the application of PAEK or composite resins might be favorable in a clinical setting, as radiographic artifacts produced by zirconia can be avoided and a differentiation of, e.g., a carious lesion from a PAEK restoration margin might be easier.

Regarding microbiological parameters, favorable results have been reported in previous investigations focusing on biofilm formation on any of the CAD/CAM materials investigated in this clinical report. For implant surfaces, zirconia showed a significant reduction in biofilm formation in comparison to titanium, while biofilm formation on PEEK was equal or lower than on zirconia or titanium [20,21]. Lower biofilm formation was identified for milled CAD/CAM composite resins in comparison to direct composite resins [22].

To date, clinical investigations focusing on tooth-colored CAD/CAM materials for multi-unit FDPs are sparse. Only short-term investigations for monolithic zirconia are available, which revealed a survival rate of 96.7% after three years [23]. The strength of this clinical report is the comparison of various contemporary CAD/CAM materials for the fabrication of 3-unit FDPs in the same prosthetic scenario. While this clinical report presented esthetical aspects as well as information on weight, radiopacity, and milling time, its results should be corroborated by clinical trials with a large number of patients using validated instruments for the patient’s esthetical perception such as the Orofacial Esthetic Scale [24].

## 5. Conclusions

Zirconia, composite resins, PEEK, and PEKK were available to fabricate 3-unit FDPs that replace a second premolar. Milling time and weight of zirconia was twice as high as for the other investigated materials. Radiopacity of zirconia restorations was intense at restoration margins and the gray values reached 255. All fabricated restorations achieved acceptable esthetics; yet, PEEK and PEKK required veneering and an opaquer was additionally used to cover the PEKK framework. However, the observations should be supported by clinical trials that use validated instruments for the estimation of esthetical perception.

## Figures and Tables

**Figure 1 materials-13-05588-f001:**
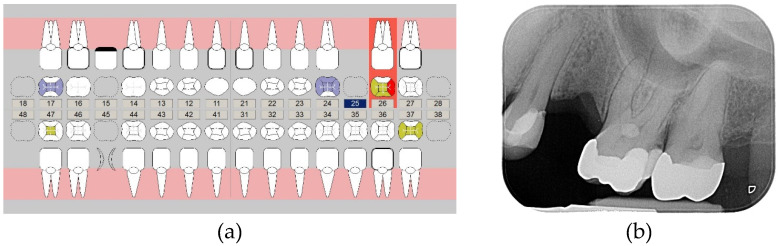
Results of the (**a**) intraoral and (**b**) X-ray examination.

**Figure 2 materials-13-05588-f002:**
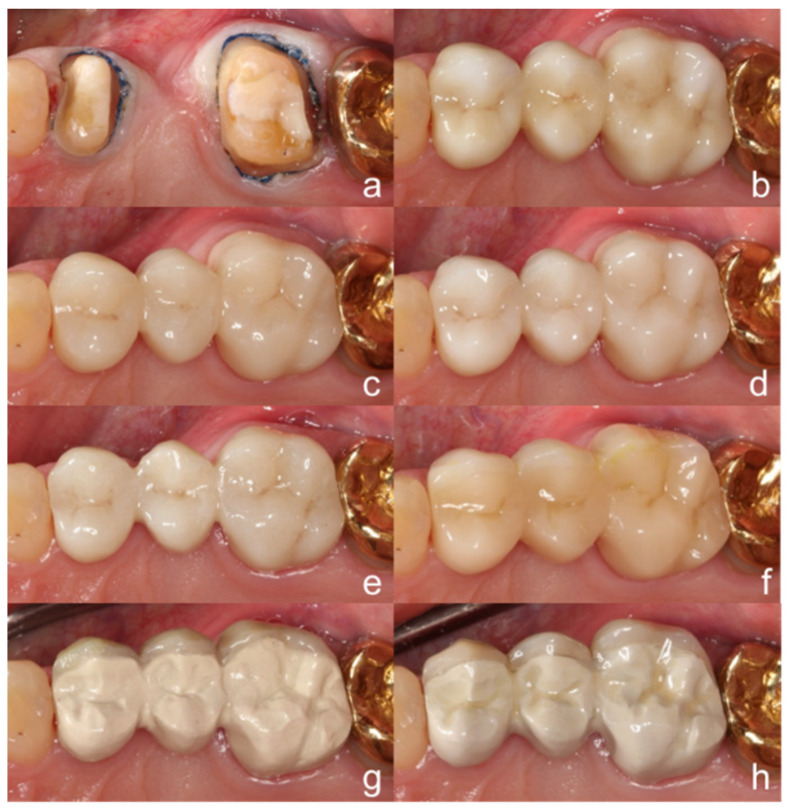
Design of the (**a**) abutment teeth and the 3-unit FDPs from an occlusal view; (**b**) 3 Y-TZP, (**c**) 4 Y-TZP, (**d**) 5 Y-TZP, (**e**) zirconia with transluency gradient, (**f**) indirect composite resin, (**g**) polyetheretherketone (PEEK), (**h**) polyetherketoneketone (PEKK).

**Figure 3 materials-13-05588-f003:**
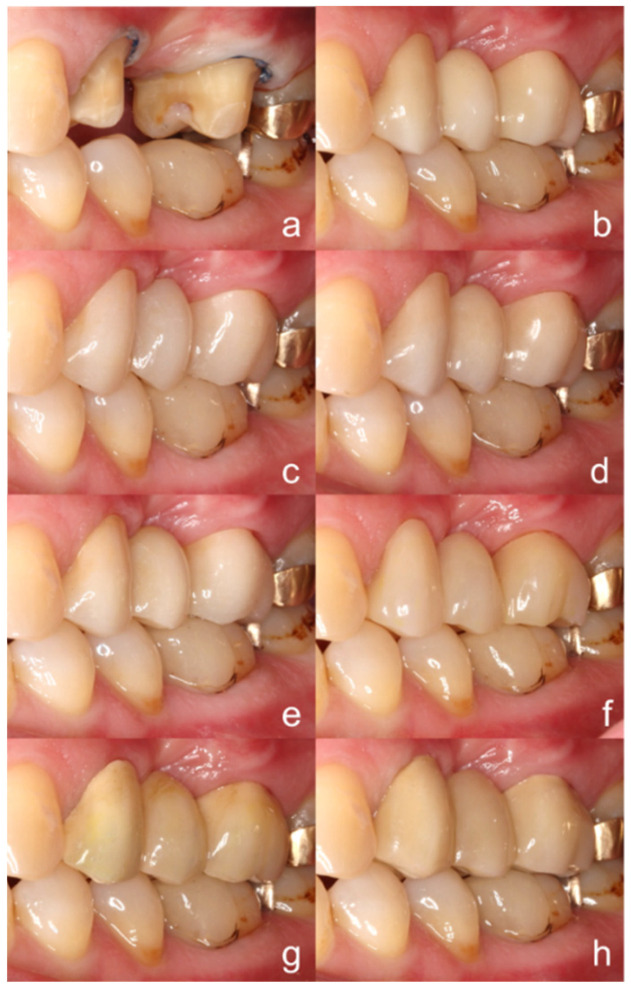
Design of the (**a**) abutment teeth and the 3-unit FDPs from a vestibular view; (**b**) 3 Y-TZP, (**c**) 4 Y-TZP, (**d**) 5 Y-TZP, (**e**) zirconia with transluency gradient, (**f**) indirect composite resin, (**g**) PEEK, (**h**) PEKK.

**Figure 4 materials-13-05588-f004:**
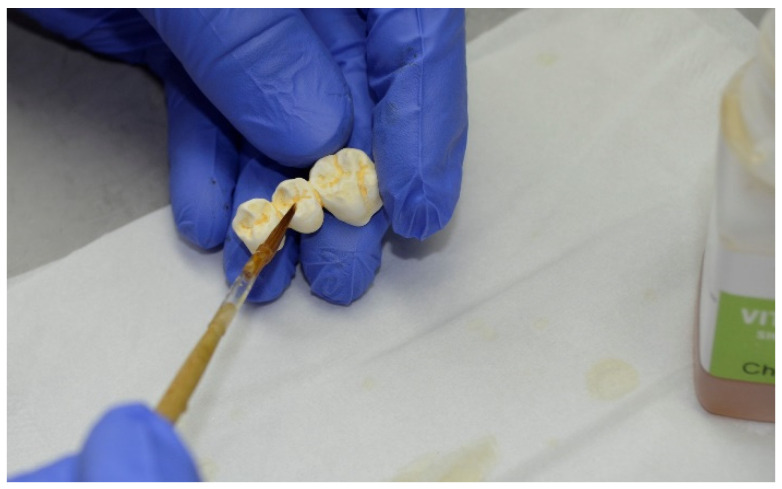
Manual coloring of the milled and white 3 Y-TZP restoration to achieve tooth color D3.

**Figure 5 materials-13-05588-f005:**
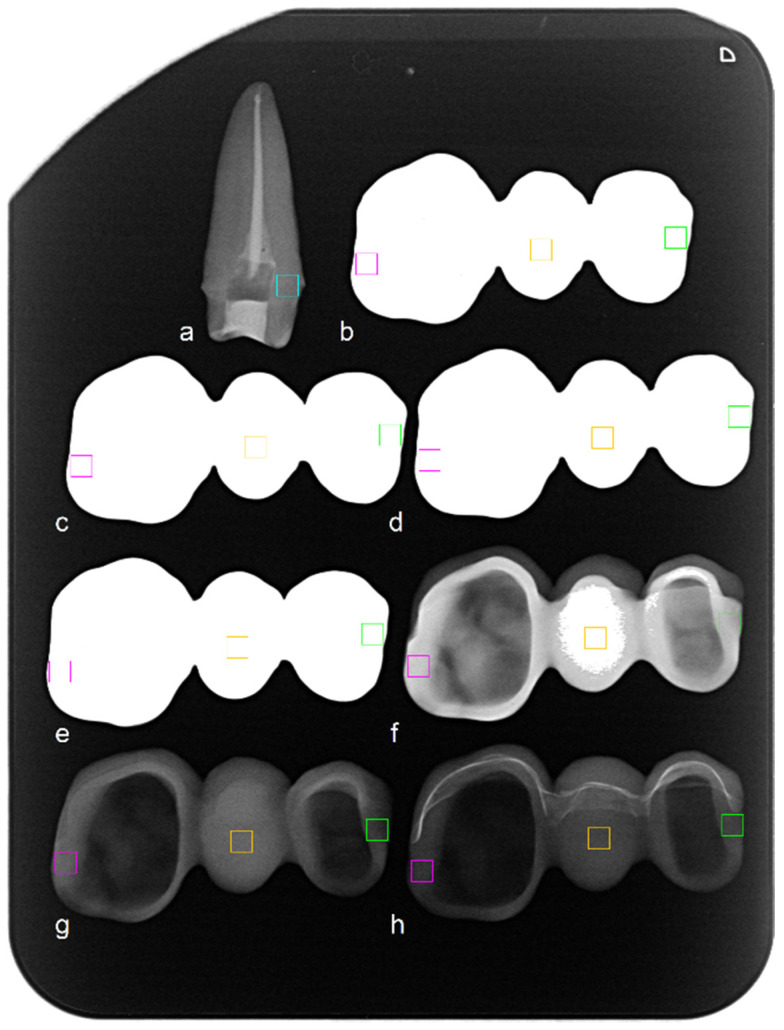
X-ray of (**a**) an extracted abutment tooth (reference) and the restorations (**b**) 3 Y-TZP, (**c**) 4 Y-TZP, (**d**) 5 Y-TZP, (**e**) zirconia with transluency gradient, (**f**) indirect composite resin, (**g**) PEEK, (**h**) PEKK with regions of interest (rectangles) for the measurements of gray values (Table 2).

**Table 1 materials-13-05588-t001:** Applied materials for the clinical report, product information according to instructions of manufacturers (Vita Zahnfabrik, Bad Säckingen, Germany; Ivoclar Vivadent, Schaan, Liechtenstein; DMG, Hamburg, Germany; Cendres Métaux, Biel, Switzerland) for posterior multi-unit fixed dental prostheses (FDPs).

Product Name, Company	VITA YZ HT ^White^, Vita Zahnfabrik	VITA YZ ST ^Multicolor^, Vita Zahnfabrik	VITA YZ XT ^Color^, Vita Zahnfabrik	IPS e.max ZirCAD Prime, Ivoclar Vivadent	LuxaCam Composite, DMG	LuxaCam PEEK, DMG	Pekkton Ivory, Cendres Métaux
Material	3 Y-TZP	4 Y-TZP	5 Y-TZP	3 and 5 Y-TZP	indirect composite resin	polyether-ether-ketone (PEEK)	polyether-ketone- ketone (PEKK)
Available colors ^a^	YZ HT ^White^ YZ HT ^Color^ (1M2, 2M2, 3M2, A1-A3)	YZ HT ^White^ YZ HT ^Color^ (A1-D4) YZ HT ^Multicolor^ (A1-D4)	YZ HT ^White^ YZ HT ^Color^ (A1-D4) YZ HT ^Multicolor^ (A1-D4)	BL1-BL4, A1-D4	Bleach 2, A1-A3, B1, C2, D2	universal	whitish
Minimum layer thickness– circular (mm)	0.5 (one pontic) 0.6 (two pontics)	0.6	1.0	1.0	0.6	0.6	0.6
Minimum layer thickness– occlusal (mm)	0.6 (one pontic) 0.7 (two pontics)	0.7 (one pontic) 0.8 (two pontics)	1.2	1.0	1.2	0.6	0.8
Minimum connector dimensions (mm^2^)	9 (3-unit FDP) 12 (>one pontic)	12 (3-unit FDP) 15 (>one pontic)	12	9 (3-unit FDP) 12 (>4-unit FDP)	16	16	14
Others	maximum of two connected pontics	maximum of two connected pontics	only 3-unit FDP, no cantilever FDP	not with bruxism, maximum of two connected pontics	only 3-unit FDP	≤4-unit FDP	not with bruxism, only 3-unit FDPs, no cantilever FDPs

^a^ please see latest product catalogue for available colors; yttria-stabilized tetragonal zirconia polycrystal ceramics (Y-TZP).

**Table 2 materials-13-05588-t002:** Features of the fabricated 3-unit FDPs to replace the second premolar in the upper jaw.

Product Name, Company	VITA YZ HT ^White^, Vita Zahnfabrik	VITA YZ ST ^Multicolor^, Vita Zahnfabrik	VITA YZ XT ^Color^, Vita Zahnfabrik	IPS e.max ZirCAD Prime, Ivoclar Vivadent	LuxaCam Composite, DMG	LuxaCam PEEK, DMG	Pekkton Ivory, Cendres Métaux
Material	3 Y-TZP	4 Y-TZP	5 Y-TZP	3 and 5 Y-TZP	indirect composite resin	Polyether-ether-ketone (PEEK)	Polyether-ketone-ketone (PEKK)
Approximated milling time (min)	51	51	51	62	32	33	34
Weight (g)	5.3	5.0	5.2	5.2	1.9	1.5	1.6
Mean gray value in X-ray ^a^							
Pontic area (orange)	255	255	255	255	255	99	67
Premolar margin (green)	255	255	255	255	174	48	62
Molar margin (pink)	255	255	255	255	199	95	47
Surface characterization	VITA akzent plus effect stains, VITA akzent plus glaze LT (both Vita Zahnfabrik)	VITA akzent plus effect stains, VITA akzent plus glaze LT (both Vita Zahnfabrik)	VITA akzent plus effect stains, VITA akzent plus glaze LT (both Vita Zahnfabrik)	VITA akzent plus effect stains, VITA akzent plus glaze LT (both Vita Zahnfabrik)	Signum Composite (Kulzer), Optiglaze Color (GC)	Signum Composite (Kulzer), Optiglaze Color (GC)	Signum Composite (Kulzer), Optiglaze Color (GC)
Others	-	-	-	-	D2 individualized with Optiglaze Color (GC) to better fit D3	-	opaquer needed to cover the framework before veneering

^a^ please see Figure 5, measured reference mean gray value revealed 114 for dentine; yttria-stabilized tetragonal zirconia polycrystal ceramics (Y-TZP).

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
