# Peer review of "Tooth-Colored CAD/CAM Materials for Application in 3-Unit Fixed Dental Prostheses in the Molar Area: An Illustrated Clinical Comparison"

_materials, 2020, doi:10.3390/ma13245588_

Round 1

Reviewer 1 Report

Dear Authors.
Congratulations on your work which, I found it very interesting. The strength of this clinical report is the comparison of various contemporary CAD/CAM materials for the fabrication of 3-unit FDPs in the same prosthetic scenario.

I have some minor revisions to propose to you to improve your work.
Please refer to the following comments.

1. Please consider changing the layout of the article

e.g.:
Introduction
Materials and Methods
Results
Discussion
Conclusions

2. The introduction should also clearly indicate the expectations towards this group of materials - indicate the advantages and disadvantages noted in the state literature.

3. The conclusions should also indicate possible ways to improve the obtained results, including aesthetic values, and a further research plan.

4. Please establish the chronology of the appearance of the pictures in the article. Now it's not in order.

Author Response

Reply to reviewer # 1

Thank you for your suggestions and helpful comments in regard to our manuscript.

  1. Concern of the reviewer: Please consider changing the layout of the article

e.g.:

Introduction

Materials and Methods

Results

Discussion

Conclusions

Our response: The structure of the manuscript was revised according to your suggestions.

Revised text: n.a.

  1. Concern of the reviewer: 2. The introduction should also clearly indicate the expectations towards this group of materials - indicate the advantages and disadvantages noted in the state literature.

Our response: The advantages and disadvantages were added to the introduction.

Revised text: please see lines 43-48 (pink), 50-51 (yellow)

  1. Concern of the reviewer: 3. The conclusions should also indicate possible ways to improve the obtained results, including aesthetic values, and a further research plan.

Our response: The conclusions were modified.

Revised text: However, the observations should be supported by clinical trials that apply validated instruments for esthetical perception.

  1. Concern of the reviewer: Please establish the chronology of the appearance of the pictures in the article. Now it's not in order.

Our response: We rearranged the chronology of pictures.

Revised text: n. a.

Reviewer 2 Report

Overall the presented clinical report is well written, and I will recommend it for publication if the issues listed below are carefully addressed:

  • Sections of the manuscript should be rearranged. The following sequence is both logical and convenient: Introduction – Materials and Methods – Results – Discussion – Conclusions.
  • Will you please specify what you meant in line “Charactherization” of Table 1. The names given in the line are names of the dyes produced by Vita Zahnfabrik. It’s unclear to me how characterization is associated with the dyes.
  • Both tables, in my opinion, belong to Materials and Methods section.
  • Below the tables there are rogue lines of text (P2L61 and P6L152). These lines seem to be fragments of sentences, but I found no corresponding sentences, where these lines belong. I suggest removing the lines.

There are also some minor language editing problems. I list here some of them.

  • I recommend replacing “relevantly” with “significantly”
  • “3nmol%”?
  • “Coincidence”. Perhaps, “coincide”
  • Remove “to” after “Besides”.

Author Response

Reply to reviewer # 2

The authors thank reviewer #2 for his/her recommendations to improve the manuscript.

  1. Concern of the reviewer: Sections of the manuscript should be rearranged. The following sequence is both logical and convenient: Introduction – Materials and Methods – Results – Discussion – Conclusions.

Our response: The manuscript was restructured.

Revised text: n.a.

  1. Concern of the reviewer: Will you please specify what you meant in line “Charactherization” of Table 1. The names given in the line are names of the dyes produced by Vita Zahnfabrik. It’s unclear to me how characterization is associated with the dyes.

Both tables, in my opinion, belong to Materials and Methods section.

Below the tables there are rogue lines of text (P2L61 and P6L152). These lines seem to be fragments of sentences, but I found no corresponding sentences, where these lines belong. I suggest removing the lines.

Our response: Thank you for the note. The name of the characterization material was wrong and was corrected by the right product name. Both tables were moved to M&M. The line fragments below the tables are footnotes that are now indicated by superscript lower-case letters.

Revised text: Please see Table 2 (green-highlighted).

  1. Concern of the reviewer: There are also some minor language editing problems. I list here some of them.

Our response: Language was revised. We would like to refrain from replacing “relevantly” by “significantly” as we believe that significantly should be used in terms of a statistical analysis.

Revised text: n.a.

Reviewer 3 Report

You can find attached comments.

Author Response

Reply to reviewer # 3

The authors thank reviewer #3 for the advices to improve the manuscript.

  1. Concern of the reviewer: General

The term ”Tooth-colored” is not well appropriate, because of some chosen biomaterials are not tooth-colored.

Our response: The authors agree with the reviewer that the materials are not available in classical tooth colors. However, the manufacturers label the colors of their materials as ‘whitish’, ‘universal’ etc., which is why we chose to compare those materials to better visualize the differences. We emphasized these aspects in the manuscript (introduction L 56-57, results L 123-125, discussion L 158-159) and hope that the reviewer agrees with our approach.

Revised text: blue-highlighted

  1. Concern of the reviewer: Introduction Milling time, weight, radiopacity and aesthetic properties were chosen and investigated. The authors must explain their choice.

Our response: We extended the explanation in the Introduction.

Revised text: Apart from potential esthetic limitations of zirconia restorations, an artifact interference during X-ray examinations was described in the scientific literature [6]. Tooth-colored CAD/CAM materials feature different levels of density and density values of 6 cm3 for zirconia could result in restorations that are heavier and need more time for milling than restorations fabricated from other CAD/CAM materials, such as composite resins or polyaryletherketones (PAEK, density 1.5 cm3) [7, 8].

  1. Concern of the reviewer: Results
  • Line 61: This sentence is linked to table 1? Its position must be revised.
  • Lines 64-65: Biomaterials densities values are enough to be able to say that zirconia is heavier that the others. No need to mill and weight FDPs to find that.
  • Lines 65-66: “The X-ray examination of the FDPs showed that zirconia restorations present intense radiographic opacity (Figure 3).” This statement is already known. This article does not add anything new to this subject. This point must be discussed and references must be added.
  • Radiopacity protocol must be improved. Radiopacity must be measured and numerical values must be used to compare biomaterials.
  • Figure 3 must be described and analysed.

Our response: The position of Line 61 was revised. We added information about the density in the introduction, however, we believe that exact values for milling time and weight are helpful to better distinguish between the materials, especially in terms of a clinical setting. The evaluation of the radiopacity was extended and Figure 5 was described and analyzed as suggested.

Revised text: L 45-48, Figure 5, Table 2, L 103-105

  1. Concern of the reviewer: Excepted some sentences of this section, the discussion is irrelevant. All this section must be revised according to the subject.

Our response: We believe that the content of the discussion is relevant to interpret the results of the current clinical report. Longevity of restorations and durability of cementation/veneering are important properties and facts that are required to correctly select from the various materials available. Knowledge about microbiological parameters is essential to favor a material in a distinct clinical setting. In addition to that, discussion about radiopacity and color were added.

  1. Concern of the reviewer: Materials and Methods
  • This part must be placed before the results section.
  • Lines 151-152: These sentences are linked to table 2? Star cannot be seen in table 2. There positions must be revised.
  • Are all FDPs fixed in the patient mouth and then removed?
  • In this section only aesthetic materials and methods is described. Weight, milling time, and radiopacity used protocols must be added in this section.

Our response: The structure of the manuscript was revised. The sentences linked to Table 2 are footnotes that are now highlighted by superscript lower-case letters. Information about the try-in procedure was added to M&M. The protocols for measurements were added.

Revised text: L 91-92, L102-106

  1. Concern of the reviewer: Conclusion

Conclusion must be improved.

Our response: The conclusion was modified.

Revised text:  L193-194; 196-197

Round 2

Reviewer 3 Report

You can find attached the file.

Author Response

The authors thank the reviewer for the time and the suggestions to improve the manuscript.

Introduction
•Milling time, weight, radiopacity and aesthetic properties were chosen and investigated. The authors must explain each choice. The link with the clinical context must be done.
What is the clinical advantage of a low milling time?
What is the clinical advantage of a light or heavy restoration?
How must be radiopacity in a clinical context?
As an example: Radiopacity is a recognized scientific property of restorative resins. It is a desirable property in that it enables the differentiation of secondary caries or decalcified dentin from the restoration, and the location of voids or other defects in the restoration, such as gingival overhanging. [A measure for quantifying the radiopacity of restorative resins. Jaideep Sur et Al. Oral Radiol (2011) 27:2227. DOI 10.1007/s11282-010-0055-4]

The authors amended the Introduction section in accordance with recommendations of the reviewer. The authors are of the opinion that weight might be relevant for patient satisfaction in cases with extensive restorative reconstructions, for instance in full-arch restorations. Milling time is rather relevant for the fabrication process in the dental laboratory than for the specific clinical setting. As a result, lower milling time may reflect simpler processing and cost-effectiveness, e.g., in a chairside approach. The authors clarified these aspects in the Introduction/Discussion section of the manuscript and hope that the reviewer agrees with this approach. A statement displaying the relevance of radiopacity has also been included as suggested.

  • Lines 45-48: values of 6cm3 and 1.5cm3 were added. Do these values concern density? Density is without unit.

The authors apologize for this mistake. The authors corrected the units to g/cm3.

Materials and methods

  • Lines 91-92: “The milling time was measured (inLab CAM SW 20.0.1, Dentsply Sirona)”
    CAM software can’t measure milling time because of the CAM software don’t take into account all the milling CNC parameters (as an example: accelerations and decelerations). The value measured by the CAM software is not the real milling time, but only an approximation.

Thank you for this comment. The authors corrected this aspect as recommended.

  • Table 2: Weight, milling time and Mean gray value in X-ray are results. These results must be moved in results section.

The authors agree with the reviewer and moved table 2 to the Results section of the manuscript.

Results

  • FDP’s weight can be compared to their density. (As they all have the same volume)
  • Radiopacity protocol must be improved. The added values are not enough to quantify radiopacity. The radiopacity values varie among the restorative materials, so one measurement is not enough.

We added information about the relation between density and weight in the Introduction and Discussion sections. We believe that exact values for the weight of the restorations are helpful for clinicians to compare the various materials. A new Figure 5 was included, too, which displays X-ray examination of the various FDPs and an extracted abutment tooth as reference. Moreover, three points of interest were defined for every restoration. Gray values were included in Table 2.

Discussion

  • In their response, authors state: “Longevity of restorations and durability of cementation/veneering are important properties and facts that are required to correctly select from the various materials available. Knowledge about microbiological parameters is essential to favor a material in a distinct clinical setting.” As ever said, discussion is not linked with the experimented parameters (Weight, radiopacity, milling time and aesthetic). All this section must be revised according to the subject.

The author thoroughly revised the Discussion section and discussed the experimental parameters in more detail. We hope that the reviewer agrees with our approach.